# Human Parsing Based Texture Transfer from Single Image to 3D Human via Cross-View Consistency

**Fang Zhao**[1]    **Shengcai Liao**[1*]    **Kaihao Zhang**[2,3]    **Ling Shao**[1,4]

[1]Inception Institute of Artificial Intelligence   [2]Australian National University
[3]Tencent AI Lab   [4]Mohamed bin Zayed University of Artificial Intelligence
{fang.zhao,shengcai.liao}@inceptioniai.org
super.khzhang@gmail.com, ling.shao@inceptioniai.org

## Abstract

This paper proposes a human parsing based texture transfer model via cross-view consistency learning to generate the texture of 3D human body from a single image. We use the semantic parsing of human body as input for providing both the shape and pose information to reduce the appearance variation of human image and preserve the spatial distribution of semantic parts. Meanwhile, in order to improve the prediction for textures of invisible parts, we explicitly enforce the consistency across different views of the same subject by exchanging the textures predicted by two views to render images during training. The perceptual loss and total variation regularization are optimized to maximize the similarity between rendered and input images, which does not necessitate extra 3D texture supervision. Experimental results on pedestrian images and fashion photos demonstrate that our method can produce higher quality textures with convincing details than other texture generation methods. Code is available at `https://github.com/zhaofang0627/HPBTT`.

## 1   Introduction

Rebuilding 3D model of human body from 2D images is of great value for many applications, such as virtual reality, movie making, clothes try-on, generation of synthetic data for learning. Particularly, generating the 3D human model from a single image has been extensively studied in recent years due to its potential practical value. However, most research works mainly focus on estimating the pose and shape of the human body [17, 31, 34, 4, 14] and very few works aim at addressing the texture generation problem.

In existing methods, [18] introduces texture inference as prediction of an image in a canonical appearance space and optimizes the perceptual metric between the rendered image and the input image. [36] proposes to generate textures of human bodies under the supervision of person re-identification (re-ID), which utilizes the distance metric learned by the re-ID task. [23] infers textures in a UV-space using an image-to-image translation method, which registers the SMPL model [26] to 3D scans of people to generate ground-truth 3D textures for training. [27] builds a paired dataset of 3D garments and 2D clothing images as training data and then learns a dense mapping from garment silhouettes to a UV map of a 3D garment model.

There still exist some issues that have not been handled well for generating textures of human body from a single image. Firstly, obtaining ground-truth 3D textures is time-consuming and labor-intensive. Secondly, textures of invisible human body parts are difficult to predict due to only one image available and lack of information from other views at inference. Thirdly, the diversity of

---

[*]Corresponding author.

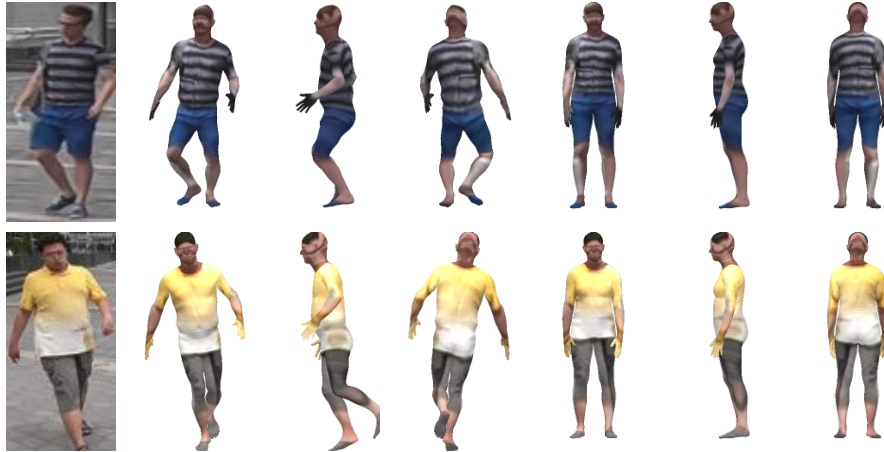

(a) Market-1501

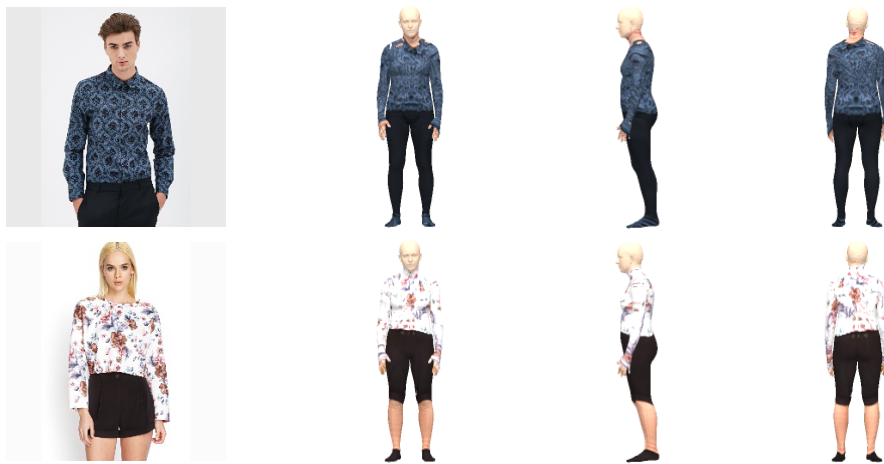

(b) DeepFashion

Figure 1: Examples of the generated textures on (a) Market-1501 [39] and (b) DeepFashion [25].

human pose and appearance makes the model hard to fit, especially when ground-truth 3D texture supervision is unavailable.

To address the aforementioned issues, we propose a human parsing based texture transfer model via cross-view consistency learning to generate the texture of 3D human body from a single image, without using 3D texture supervision. Examples of the generated textures are shown in Fig. 1. We first use the semantic parsing of human body as the model input. Compared to image pixels and silhouettes, human parsing reduces the appearance variation of human image and preserves its pose information. Then, an encoder employs two Convolutional Neural Networks (CNN) to extract shape and pose features from the human parsing, respectively. After that, a decoder combines features of shape and pose and deconvolves to produce a texture flow, which stores coordinates of the input image to sample pixel values of a texture image from. In order to improve the texture prediction for invisible parts of human body, we explicitly enforce the cross-view consistency of texture prediction between two images with different views of the same subject during training. Specifically, the texture predicted by one view is used to render with the 3D mesh of another view and enforced to match the input image of another view. Finally, we optimize the perceptual loss and total variation regularization to maximize the similarity between rendered and input images.

Our main contributions include the following three aspects: 1) We propose a novel texture transfer model to effectively generate textures of 3D human body from a single image via cross-view consistency learning. 2) We leverage the semantic parsing of human body as input to reduce the human appearance variation and preserve the pose information. 3) Our model produces the state of

the art quality of textures on both pedestrian images from the surveillance scene and fashion photos from the web.

## 2 Related Work

**Texture generation of 3D model.** The texture generation of 3D model produces textures on 3D model surface given one or multiple 2D images. These works mostly focus on combining texture pieces from multiple images. Some methods [24, 12, 35] try to mitigate the problems of seams and broken textures while mapping photographs onto appropriate 3D mesh surface. For example, [12] assigns compatible texture to adjacent triangles by searching combinatorially over the source images and a set of local image transformations that compensate for geometric misalignment. Some other methods [9, 20] fuse multiple images to generate textures by designing various weighted average strategies. There are also some methods [10, 41, 7, 2, 3] which are based on warping refinement. These methods usually need images with different views or RGB-D sensors to infer 3D textures. [41] maps color images onto geometric reconstructions by optimizing camera poses in tandem with non-rigid correction functions for all images to maximize the photometric consistency of the reconstructed mapping. [7] proposes a global patch-based optimization framework to synthesize the aligned images, which uses patch-based synthesis to reconstruct a set of photometrically-consistent aligned images by drawing information from the source images. [2] stitches a complete texture using graph-cut based optimization based on a semantic texture prior. [3] warps the estimated canonical model back to each frame and back-projects the image color to all visible vertices to generates a texture image. Besides, as mentioned in Section 1, only very few works [18, 36, 23, 27] are proposed to address the problem of generating textures from a single image. The most related work to ours is [36] which also uses a pre-trained human body mesh model and a differentiable renderer. Different from [36] , we explicitly encourage the cross-view consistency during training. Moreover, we propose to use the semantic parsing of human body as the model input to reduce the appearance variation of human image and preserve its pose information.

**3D human reconstruction.** The 3D human reconstruction aims to reconstruct 3D human shape and pose under a specific body model. Some methods [6, 13] use key-points and silhouettes of human body to estimate the pose and shape parameters of SCAPE [5], which is a data-driven human shape model. Some of recent methods adopt more powerful SMPL [26] as their human body model. SMPL is a skinned vertex-based model which can represent a wide variety of body shapes in natural human poses. [8] learns the SMPL model by minimizing the difference between projected 3D body joints and detected 2D joints and preventing the inter-penetration between limbs and trunk. [22] accelerates the SMPL model by inferring the 3D shape and pose directly from 91 landmark predictions in UP-3D dataset. [17] iteratively regresses the SMPL parameters by generative adversarial network to learn more real human shape and pose. [30] improves human mesh estimation using multi-view textures as a weak-supervision signal. There also exist some methods that are built on more complicated models with more body details [3, 1, 4, 11, 29, 33, 14]. [3] proposes to transform the silhouette cones corresponding to dynamic human silhouettes to obtain a visual hull in a common reference frame. [1] predicts shape in a canonical T-pose space using both bottom-up and top-down streams allowing information to flow in both directions. [4] turns shape regression into an aligned image-to-image translation problem and estimates detailed normal and vector displacement maps from a partial texture. [14] learns two separate networks that disentangle the task into a pose estimation and a non-rigid surface deformation step in a weakly supervised manner. [32] introduces a self-supervised approach to learn a powerful representation for 3D pose estimation, where a spatial transform based bidirectional novel view synthesis is further proposed to exploit view consistency. In contrast, our work is to generate a physical 3D model for computer graphics system, with a parameterized 3D mesh along with detailed textures on it.

## 3 Method

We aim to learn a predictor that can infer the texture of the 3D human body from a single image without using 3D ground-truth textures. In this section, we present the proposed texture transfer model consisting of an encoder which extracts shape and pose information from the semantic parsing of human body, and a decoder which predicts the texture flow by combining shape and pose features.

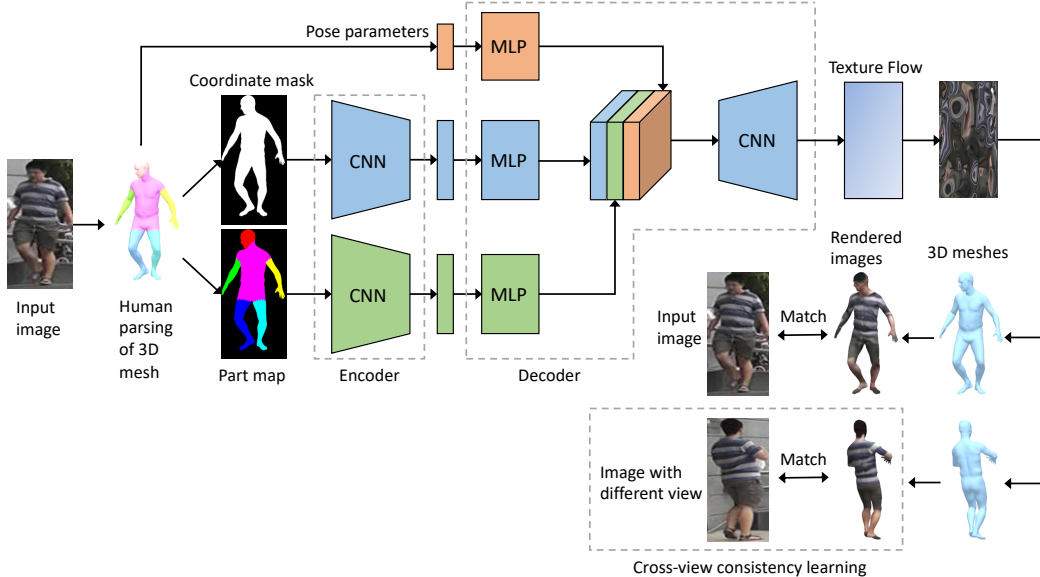

Figure 2: Overview of the proposed texture transfer model via cross-view consistency learning. The proposed model contains an encoder which extracts shape and pose information from the semantic parsing of human body and a decoder which predicts the texture flow by combining shape and pose features. A cross-view consistency learning strategy is introduced to enforce the rendered image to match the image with different view.

We also propose a cross-view consistency learning strategy to enforce the rendered image to match the image with different view. The overall framework is illustrated in Fig. 2.

## 3.1 3D Human Body Model

We represent the 3D human body model as a 3D mesh $M$, which is defined by vertices $\mathbf{V} \in \mathbb{R}^{|\mathbf{V}| \times 3}$ and faces, and then parameterize it with the SMPL body model [26] as $M(\beta, \theta, \gamma)$ controlled by shape parameters $\beta$, pose parameters $\theta$ and translation parameters $\gamma$. To reduce the impact of various body shapes, poses and translations during texture prediction learning, we employ HMR [17], which is the state-of-the-art method of 3D human body shape and pose estimation, to obtain parameters of the shape, pose and translation of the input human image. Then, we use these parameters to align the rendered human image with the input image and keep them fixed in the follow-up training.

## 3.2 Texture Prediction with Human Parsing

Once we get the 3D mesh $M(\beta, \theta, \gamma)$ of human body, we can learn the texture transfer from the image to the 3D mesh surface. However, directly predicting textures in 3D space is difficult. Here we first compute a 2D texture image $\mathbf{I}^{uv}$ with height $H_{uv}$ and width $W_{uv}$ and then project the image $\mathbf{I}^{uv}$ to the mesh surface via a fixed UV mapping [16]. In order to preserve more details of textures, similar to [18], instead of regressing the pixel values of $\mathbf{I}^{uv}$, we predict the texture flow $\mathbf{F} \in \mathbb{R}^{H_{uv} \times W_{uv} \times 2}$, which denotes the coordinates of the input image to sample the pixel values of $\mathbf{I}^{uv}$ from. After that, the bilinear sampling $g$ is applied on the input image $\mathbf{I}$ by using the predicted flow $\mathbf{F}$ to obtain the texture image: $\mathbf{I}^{uv} = g(\mathbf{I}; \mathbf{F})$.

Our texture transfer model first employs an encoder to extract features of the input image. Because the location where the input image pixel is transferred to the 3D mesh is irrelevant with its value, but only depends on its location on the input image, we use the semantic parsing of human body as the input of the encoder instead of the original image. Here the human parsing can be obtained easily by using part labels of the SMPL body mesh [22], without the aid of any external algorithm. Different from [28], where clothing image silhouettes are used as the input and those clothing images from the online shop have good pre-segmentation and less pose variation, the human images considered in our task have multiple semantic parts and complicated poses. Thus, we propose to leverage the human parsing to provide the information of human part and pose besides the silhouette shape.

As shown in Fig. 2, we segment the body mesh into 6 parts, including *head*, *torso*, *left-arm*, *right-arm*, *left-leg* and *right-leg*. We further render the human parsing of the body mesh to get two components. One is a coordinate mask $\mathbf{I}^{mask} \in \mathbb{R}^{H \times W \times 2}$, which stores at each image pixel location its own coordinates if the pixel belongs to the human body, and 0 otherwise [28]. The coordinate mask $\mathbf{I}^{mask}$ provides the shape information of human body. The other is a semantic part map $\mathbf{I}^{part} \in \mathbb{R}^{H \times W \times 3}$, which represents the semantic parts by using 6 sets of RGB values. The part map $\mathbf{I}^{part}$ indicates the spatial distribution of semantic parts and pro-

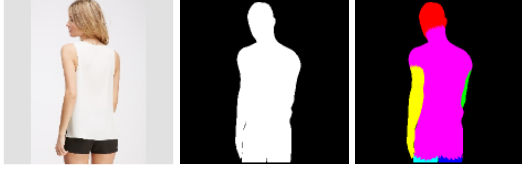

Figure 3: Motivation of using the human parsing as the input. With the human silhouette alone, it is hard to know the arm position and body orientation. However, these information can be easily obtained from the human parsing.

vides the pose information of human body. Fig. 3 shows that it is crucial for inferring accurate textures. Obviously, with the human silhouette alone, it is hard to know the arm position and body orientation. However, these information can be easily obtained from the human parsing. Our encoder uses two Convolutional Neural Networks (CNN) respectively to embed $\mathbf{I}^{mask}$ and $\mathbf{I}^{part}$ into a common feature space: $f_{enc}^1(\mathbf{I}^{mask}) \in \mathbb{R}^K$, $f_{enc}^2(\mathbf{I}^{part}) \in \mathbb{R}^K$.

Once we extract the features of $\mathbf{I}^{mask}$ and $\mathbf{I}^{part}$, we employ a decoder to combine the features and the pose parameters $\theta$ obtained by HMR, which provides additional pose information encoded in SMPL, and deconvolve them to produce the texture flow $\mathbf{F}$:

$$\mathbf{F} = f_{dec}([f_{enc}^1(\mathbf{I}^{mask}), f_{enc}^2(\mathbf{I}^{part}), \theta]). \tag{1}$$

Our decoder first uses Multilayer Perceptrons (MLP) to expand the input vectors to 3D tensors and concatenates them along the channel dimension, and then feeds the combined feature map into a series of deconvolutional layers.

Finally, we adopt NMR [19], a differentiable renderer, to render the human body mesh $M$ with the predicted texture image $\mathbf{I}^{uv}$ to obtain the rendered image:

$$\mathbf{I}^{rend} = R(M, \mathbf{I}^{uv}). \tag{2}$$

### 3.3 Model Learning via Cross-View Consistency

In order to learn our texture transfer model, the perceptual loss [38] is optimized to make the rendered image $\mathbf{I}^{rend}$ similar with the input image $\mathbf{I}$. Since only one image is available for the inference, some human parts are invisible, which may affect the texture prediction for these parts. Thus, we explicitly enforce the consistency of predicted textures across different views of the same subject during training. Specifically, we select two images $\mathbf{I}_1$ and $\mathbf{I}_2$ with different views from the same subject and estimate their SMPL parameters using HMR to obtain their 3D body mesh $M_1$ and $M_2$. Through our encoder and decoder, we can have the predicted texture images $\mathbf{I}_1^{uv}$ and $\mathbf{I}_2^{uv}$. Then, we use $\mathbf{I}_1^{uv}$ to render $M_2$ and $\mathbf{I}_2^{uv}$ to render $M_1$:

$$\mathbf{I}_{1 \to 2}^{rend} = R(M_2, \mathbf{I}_1^{uv}), \; \mathbf{I}_{2 \to 1}^{rend} = R(M_1, \mathbf{I}_2^{uv}). \tag{3}$$

Intuitively, because $\mathbf{I}_1^{uv}$ and $\mathbf{I}_2^{uv}$ come from the same subject, the rendered images $\mathbf{I}_{1 \to 2}^{rend}$ and $\mathbf{I}_{2 \to 1}^{rend}$ should be consistent with the input images $\mathbf{I}_2$ and $\mathbf{I}_1$, respectively. Thus, we learn the model by not only matching the rendered images with its own input image but also the input image of another view. The texture transfer loss is given by

$$L_{tex} = \sum_n L_{perc}(\mathbf{I}_n^{rend}, \mathbf{I}_n) + \sum_{n,k} L_{perc}(\mathbf{I}_{n \to k}^{rend}, \mathbf{I}_k) + L_{perc}(\mathbf{I}_{k \to n}^{rend}, \mathbf{I}_n). \tag{4}$$

We also use an anisotropic version of the total variation regularization to make the texture flow $\mathbf{F}$ produced by the decoder remain smooth.

$$L_{tv} = \sum_{i,j,c} |F_{i+1,j,c} - F_{i,j,c}| + |F_{i,j+1,c} - F_{i,j,c}|. \tag{5}$$

Thus, the overall objective function is

$$L = L_{tex} + \lambda L_{tv}, \tag{6}$$

where $\lambda$ is the weight of the total variation regularization.

# 4 Experiments

We evaluate the proposed texture transfer model on two human datasets, i.e., Market-1501 [39] and DeepFashion [25], which contain human images with various views and poses. Following [36], we adopt the Structural Similarity (SSIM) [37] and its masked version mask-SSIM to measure the quality of generated textures. We also compare the proposed model against the related texture generation methods [18, 36].

## 4.1 Datasets

Market-1501 [39] is a person re-identification dataset, which consists of 32,668 images of 1,501 persons from six disjoint surveillance cameras. The image size is $128 \times 64$. We regard images with the same person identity as belonging to the same subject. 12,936 images of 751 identities are used for training. 3,368 query images from the remaining 750 identities and 19,732 gallery images form the test set. In our experiments, we use the query set for testing.

DeepFashion (In-shop Clothes Retrieval Benchmark) [25] includes 52,712 in-shop clothes images with large pose, view and scale variations. All images have size $256 \times 256$. We regard images with the same clothes as belonging to the same subject. After removing images only containing a small part of human body, we use 20,185 images from the original training set for training, 6,639 images from the original query set for testing.

## 4.2 Implementation Details

In all experiments, we use the Adam [21] optimizer with $\beta_1 = 0.9$ and $\beta_2 = 0.999$. The learning rate is $1 \times 10^{-4}$. The model is trained with the mini-batch size 16 for 93k iterations on Market-1501 and 100k iterations on DeepFashion. In order to enable the cross-view learning, each mini-batch consists of 8 subjects and each subject has two images. The weight $\lambda$ of the total variation regularization is set empirically to 0.5. Similar to [36], for Market-1501, background images are randomly cropped from the PRW dataset [40] and added to the rendered images, and for DeepFashion, the texture of face part is fixed because here we mainly consider the body texture generation.

For the network architecture, the encoder adopts two ResNet-18 [15] with random initialization to extract 4096-D feature vectors of the coordinate mask and semantic part map, respectively, and feeds them into two fully-connected layers to get 200-D feature vectors. The decoder adopts 3 two-layer MLP to combine the encoded features and SMPL pose parameters, and then passes the combined feature map through 6 deconvolutional layers. At last, a tanh activation function is applied to normalize the texture flow to $[-1, 1]$.

## 4.3 Ablation Study

We firstly verify the effectiveness of main components in our proposed texture transfer model by both qualitative and quantitative evaluations on the Market-1501 dataset.

**Semantic part map.** To investigate the impact of our semantic part map, we only use the coordinate mask as the model input and show images rendered with the predicted textures in the column "No-PM" of Fig. 4. As one can see, if the semantic part map is not employed, some small body parts of the rendered images, e.g., arms, hands and legs, are not textured appropriately. This indicates the semantic part map can provide the information of spatial distribution of body parts for the model. We also report the SSIM and mask-SSIM scores in Table. 1. The model without using the semantic part map obtains worse scores, which is consistent with the qualitative results.

**Cross-view consistency.** We only use single view image matching during training to evaluate the influence of the proposed cross-view consistency learning. The results are illustrated in the column "No-CC" of Fig. 4. It can be observed that the textures of some invisible parts in the input images cannot be inferred correctly, such as profiles and occluded logos, which shows enforcing the consistency between rendered images with different views during training is able to make the learned model more robust for invisible body parts. The quantitative results are also reported in Table. 1. The model without cross-view consistency learning obtains 0.305 and 0.863 for the SSIM and mask-SSIM scores respectively, which are lower than our model.

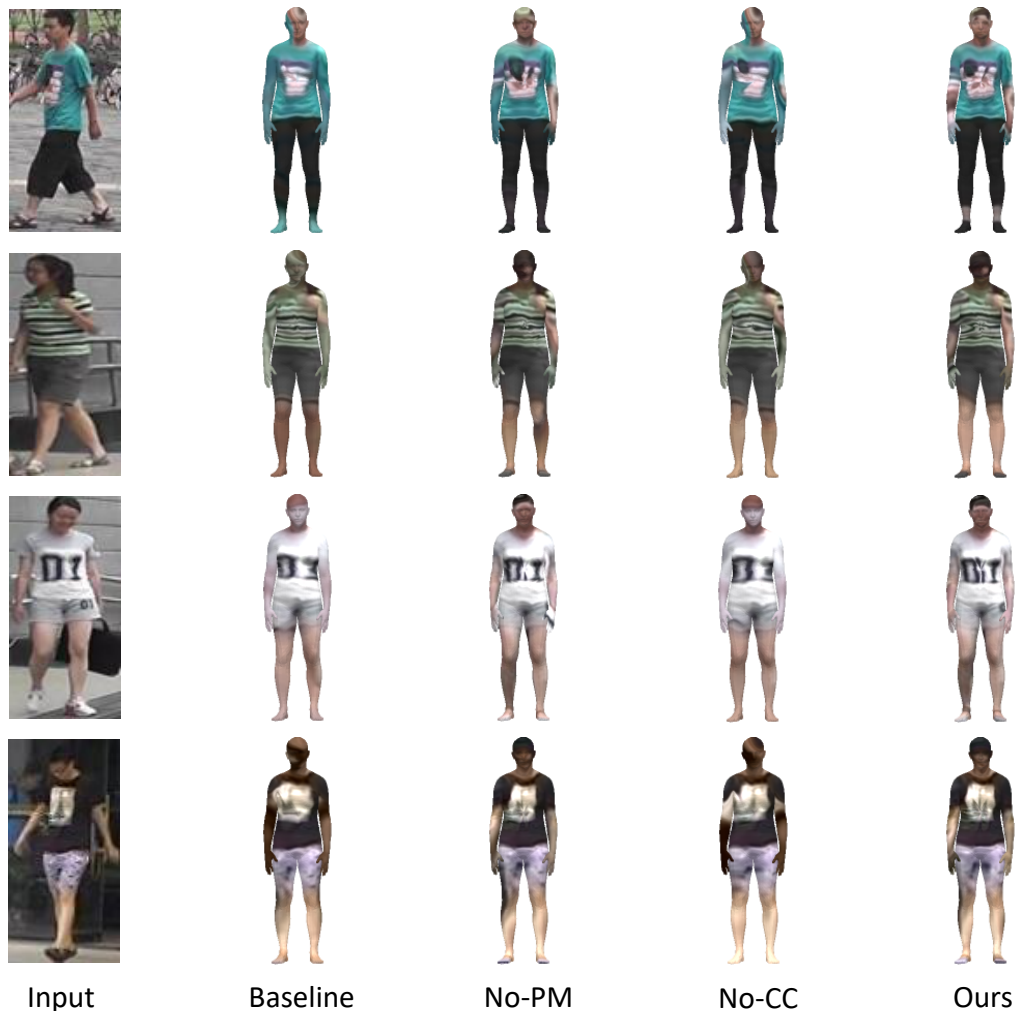

| Input | Baseline | No-PM | No-CC | Ours |

Figure 4: Textured human bodies predicted by different components in our model on Market-1501, including the baseline (CMR [18]), No-PM (without semantic part map), No-CC (without cross-view consistency) and our model.

Table 1: SSIM and mask-SSIM scores obtained by different components in our model on Market-1501, including the baseline (CMR [18] ), No-PM (without semantic part map), No-Pose (without SMPL pose), No-CC (without cross-view consistency), No-TV (without total variation regularization) and our model.

| Model | Baseline | No-PM | No-Pose | No-CC | No-TV | Ours |
|---|---|---|---|---|---|---|
| SSIM | 0.276 | 0.310 | 0.315 | 0.305 | 0.316 | **0.318** |
| mask-SSIM | 0.836 | 0.869 | 0.873 | 0.863 | 0.875 | **0.877** |

**SMPL pose.** We employ the pose parameters of the SMPL body model as a complement to the semantic part map. As shown in Table. 1, the SSIM and mask-SSIM scores decline when removing the SMPL pose from our model, which indicates that the SMPL pose can provide extra pose information encoded in SMPL for the model learning.

**Total variation regularization.** We also use the total variation regularization to smooth the predicted texture flow during training. From Table. 1, one can see that the performance slightly drops if without using the total variation regularization because it can guarantee the neighborhood correlation of predicted coordinates in the input image.

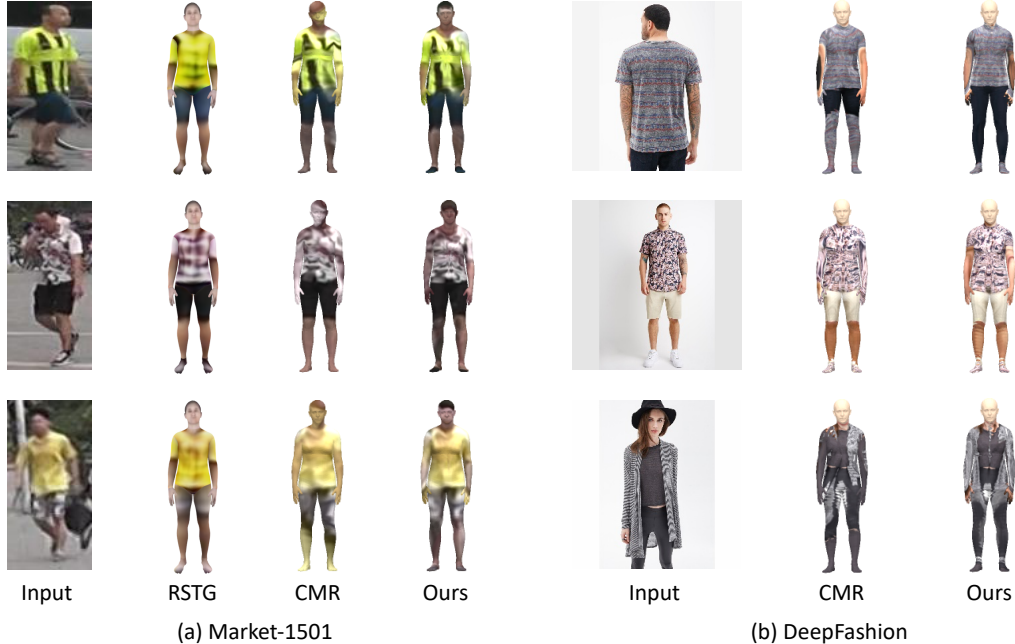

| Input | RSTG | CMR | Ours | Input | CMR | Ours |

(a) Market-1501            (b) DeepFashion

Figure 5: Textured human bodies predicted by different texture generation methods on (a) Market-1501 and (b) DeepFashion.

### 4.4 Comparison with Related Methods

We now compare the proposed texture transfer model against the related texture generation methods, including CMR [18] and RSTG [36] on the Market-1501 and DeepFashion datasets. Fig. 5 and Fig. 6 illustrate the textures predicted by different methods and Table 2 lists their SSIM and mask-SSIM scores.

As shown in Fig. 5, our model recovers the texture of 3D human body with more details while keeping accurate semantic part correspondence with 2D image. RSTG loses too many texture details although it predicts the semantic part distribution of textures on the 3D mesh well. For example, on the first row of Fig. 5 (a), black strips on the shirt in the input image are completely lost in the texture generated by RSTG. The reason is that RSTG directly predicts the pixel values of textures, which cannot preserve the texture information of the input image very well compared with the texture flow. CMR is our baseline, which uses the image pixel values as input to predict the texture flow with single view learning. From Fig. 5 one can see that CMR cannot preserve the semantic part correspondence precisely and sometimes loses parts of textures because the appearance variation of input image weakens the robustness of the model. Besides, CMR cannot handle well the texture prediction for invisible parts, e.g., the legs in the input image on the first row of Fig. 5 (b). Instead, our model is able to infer the textures of those parts successfully by the cross-view consistency learning. We also show comparisons from multiple views of textured human bodies in Fig. 6. It can be seen that our model still obtains consistently higher quality textures under different views.

For the quantitative results, our model achieves the highest SSIM and mask-SSIM scores on both Market-1501 and DeepFashion, as reported in Table. 2. This further demonstrates the effectiveness of the proposed human parsing based texture transfer and cross-view consistency learning.

## 5 Conclusion and Future Work

In this paper, we propose the human parsing based texture transfer model via cross-view consistency learning to generate the texture of 3D human body from a single image. The semantic parsing of human body is used as the model input, which provides both the shape and pose information, to reduce the appearance variation of human image and preserve the spatial distribution of semantic parts. To improve the prediction for textures of invisible parts, the consistency across different views

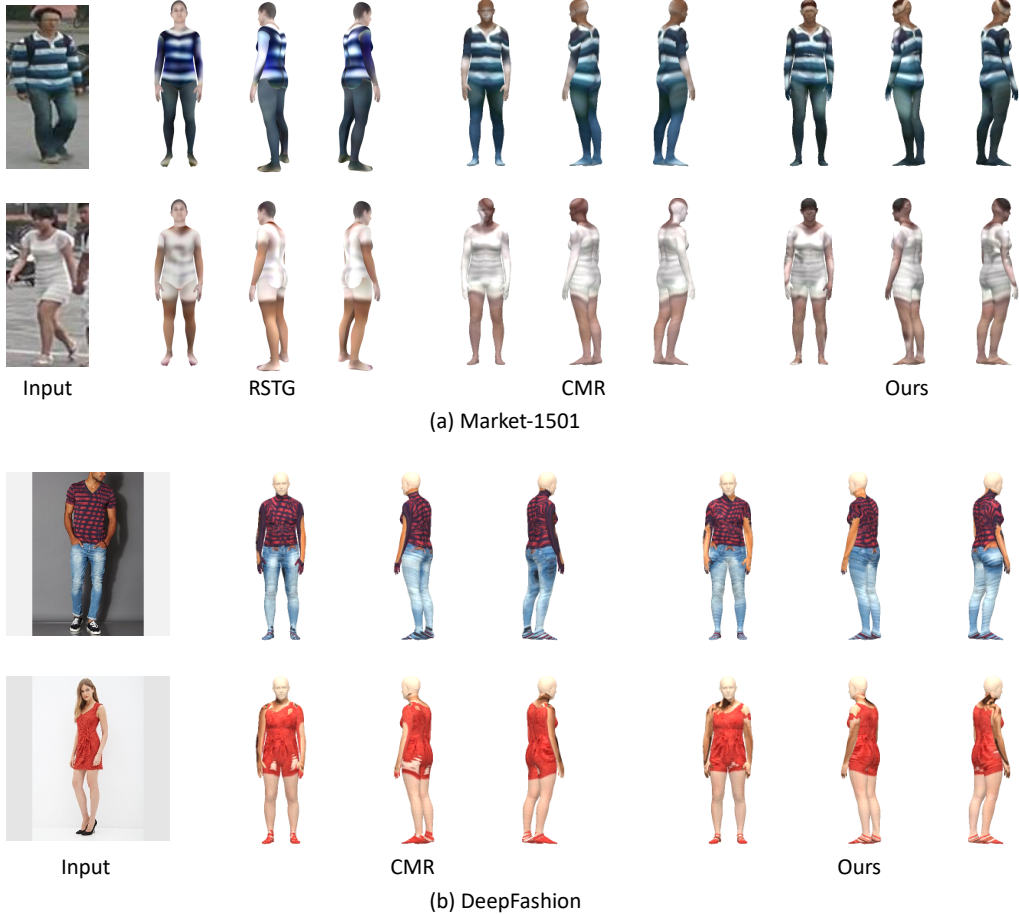

Figure 6: Comparisons from multiple views of textured human bodies on (a) Market-1501 and (b) DeepFashion.

Table 2: SSIM and mask-SSIM scores obtained by different texture generation methods.

| Method | Market-1501 | | DeepFashion |
|---|---|---|---|
| | SSIM | mask-SSIM | SSIM |
| CMR [18] | 0.276 | 0.836 | 0.709 |
| RSTG [36] | 0.164 | 0.372 | - |
| Ours | **0.318** | **0.877** | **0.735** |

of the same subject is learned by not only matching the rendered images with its own input image but also the input image of another view. Experimental results on two human datasets demonstrate that our method is able to generate textures with convincing details for 3D human body.

Future work could focus on improving texture prediction for patterns totally lost in input images, e.g., patterns only in the front or back view, in this case the current model tends to simply copy the textures from the back or front view. Since only one image is used as input at inference and ground-truth 3D textures are unavailable during training, it is very hard to infer such textures, especially for texture flow prediction which needs to sample pixel values in the input image. Further improvements could be possible with prior knowledge about front-back texture relations learned from data.

# Broader Impact

Enabling machine learning models to understand and reconstruct 3D data is able to significantly improve the experience of human-machine interaction, such as virtual reality, clothes try-on and so on. Besides, it also facilitates the invariant and robust representation learning from the geometry effects, such as scales and views. Texture generation of 3D model is a critical part in 3D reconstruction. Particularly, the texture generation model based on a single image can extremely reduce the resources required by model learning and inference, and shows great potential for industrial applications. However, poses, shapes and textures estimated by the model could be abused to synthesize fake pictures of people, which is a negative aspect.

# Acknowledgments

This work was supported in part by the National Natural Science Foundation of China (NSFC) Project #61672521.

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
