[Reviews · NeurIPS 2020]

Review 1

Summary and Contributions: This paper proposes cross-view consistency to predict texture from a single image, without using any ground truth 3D texture. It builds upon the texture flow proposed in CMR, augmented with part segmentation obtained via 3D mesh reconstruction. The idea is that a single texture must explain the image seen from another view. Results compare with two prior method that can be used to recover texture in without explicit 3D or texture information.

Strengths: - The proposed signal is cross-view, not multiview (taken at the same time), via identity maching over a video. While there are several papers that learn texture through multiview signal, it's interesting to do this in a cross-view setup. This is less intensive than multiview signal which requires a lot more setup to capture. While [33] also utilizes re-id information and therefore implicitly cross-view, this instantiation is more explicit. - Good ablation. - Transfer experiments on DeepFashion is a good effort, it does seem to transfer to it which is promising.

Weaknesses: - It's unclear if there is significant improvement over RSTG[33] from Figure 5. In particular, the results are only compared from the frontal view, the approach should be compared with [33] that shows multiple views of the image. The results of [33] is not compared on DeepFashion. the SSIM metric does show different, but SSIMs are not very good measurement for these types of evaluations. I recommend using LPIPS. In fact, CMR looks a lot worse perceptually than RSTG in Figure 5(a), however there is a significant difference in mask-SSIM which is a bit peculiar. - The approach looks like it's using a very simple spherical UV mapping as was done in CMR. For human body shpaes, the simple spherical UV mapping introduces quite a significant distortion. It's quite remarkable that the network can overcome this. I suggest using the disjoint UV mapping that is used for humans (as shown in [23]), and texture flow prediction on disjoint uv maps have been done in "Three-D Safari: Learning to Estimate Zebra Pose, Shape, and Texture from Images "In the Wild" Zuffi et al. ICCV '19. or that use din [33]. This may improve the results. - The particular instantiation of cross-view consistency may be novel, however all the pieces have been explored before. RTSG [33] also uses re-id to supervise the texture. Missing relevant citations: TexturePose: Supervising Human Mesh Estimation with Texture Consistency, Pavlakos et al. ICCV'19 SiCloPe: Silhouette-Based Clothed People, Natume et al. CVPR '19 PIFu: Pixel-Aligned Implicit Function for High-Resolution Clothed Human Digitization, Saito et al. ICCV '19 SiClope and PIfu both the texture of unseen regions and tested on DeepFashion. These approaches are supervised, but they could be cited. TexturePose is a relevant in that it uses multi-view texture as a weak-supervision signal to improve the pose prediction. - No supplemental, few qualitative results, and only few qualitative results are shown in from multiple views (Figure 1)

Correctness: Yes, however I recommend using a better UV mapping than the spherical coordinates. The spherical UV mapping does not make much sense for human shapes as described above.

Clarity: yes.

Relation to Prior Work: The paper should discuss the difference wrt [33] more clearly in the related work section. Currently it only appears as one of the lists and not much discussion is had around this, even though it uses a similar cue and compared in the evaluation.

Reproducibility: Yes

Additional Feedback: I think this paper is a good effort, but builds upon existing components, and most significantly it's unclear if there is much improvements upon [33], which also uses a similar, albeit different instantiation of, weak supervisory signal . Further more, the results are on the underwhelming side. This is possible because it is trained on low-resolution images from Market-1501 without any ground truth supervision, and the recent results of _supervised_ methods on DeepFashion (Siclope, Pifu et al) are very impressive. However, the results do not seem significant enough with previous work to warrant a publication at NeurIPS. === Update After the response + discussion, the difference with RSTG is clarified and I am happy to raise my score to 6.


Review 2

Summary and Contributions: This paper proposes a novel method for single RGB texture reconstruction, with NMR-based cross-view consistency applied to the same subject with different viewpoints in the dataset. It also leverages the information of human semantic segmentation to improve texture-flow accuracy.

Strengths: This paper proposes a novel framework for single-view human texture reconstruction with cross-view accuracy during the training phase. It leverages the texture-flow structure to directly obtain texture information from the image, and the use of parsing information improves the quality of texture-flow. The method of cross-view "cross-flow" rendering using NMR to guarantee cross-view consistency is effective and well evaluated. The ablation study is sufficient and the qualitative and quantitative evaluation and comparison is well performed.

Weaknesses: 1. The quality of the texture is not very satisfying, in Fig.4, there are some artifacts on the arm area (line 2) and even on the front-view (line 3). Also, the results show that the texture copying from the image a little "randomly" when the texture is complicated (Fig.1 DeepFashion part), the results shows few texture details and patterns. 2. It seems that the texture tends to have front-back symmetry in Fig.1. Will the texture have similar phenomena in other data, e.g., when there is a big logo on the front, will the generated texture have a logo on the back? I don't see back-view results with logo on the front. There should be more discussion on this phenomena and more back-view results especially with the above conditions. 3. The comparison with RSTG[33] should also add results rendered in other views, in order to better claim the effectiveness of the cross-view loss. 4. More comparison can be established, e.g. with PIFu.

Correctness: The claims and methodis are correct and well evaluated.

Clarity: The paper is well written with clear language and structure.

Relation to Prior Work: The paper clearly discussed its relation with previous work, and make qualititative and quantitative comparison.

Reproducibility: Yes

Additional Feedback: === Update After reading the response from the author and the reviews & discussions from other reviewers, I would like to maintain my score as 6.


Review 3

Summary and Contributions: The authors propose a method for generating textures of worn clothing from a single image, including that of occluded parts. Key components are a warping-based approach that establishes a spatial correspondence between the image and texture coordinates (as used elsewhere) and a cross-view consistency term (appears new).

Strengths: Using human pose information from the target view to establish correspondences is simple yet efficient. It is related to the bidirectional novel-view synthesis proposed in [A], which should be discussed and differences highlighted. Since this appears to be the main contribution of this paper, what are the closest exiting methods to this strategy? Certainly there are novel-view synthesis works, but are there ones using detections in the target view to establish pixel correspondence over views?

Weaknesses: Using a pre-trained method for 3D pose and shape estimation accumulates its errors. The method would be much stronger if the pre-trained model would onlye be used for bootstrapping. Could the existing model be trained end-to-end, including pose and shape estimation? It would be a nice direction for future work / future iterations of this work.

Correctness: Limitations should be stated more clearly. E.g. what happens when the pose or shape estimation is inaccurate? Some artifacts can be seen when the arm occludes the body. Moreover, why is the head excluded in DeepFashion? The method is sound, but I'm not 100% sure if there isn't related work using the same view consistency. The method is evaluated on two established and very different datasets.

Clarity: The paper is well-written and easy to understand. The part about how correspondence between two views is established could be explained in more prominence and discussed in relation to existing methods, such as [A]. The broader impact session should also discuss negative aspects, e.g. ways pose and shape estimation could be abused.

Relation to Prior Work: View consistency by using detections in a targe view have been used in other form in: [A] Rhoding et al. Neural Scene Decomposition for Multi-Person Motion Capture. CVPR 2019

Reproducibility: Yes

Additional Feedback: I would not call the poses in deep fashion that diverse, they are all standing with typical posing poses. Thanks for the rebuttal, I stand to my score.

[Author Response · NeurIPS 2020]

We appreciate all reviewers' valuable comments. We shall address the concerns raised point by point, as follows.

**To R1 & R2: 1. The qualitative results are not very satisfying:** In fact, there is very little supervision used in our method and so it is very challenging. Firstly, we do not use ground-truth 3D texture supervision. Second, the human parsing, body shape and pose are directly obtained from the HMR [17] model pre-trained on other datasets, which may cause inaccurate estimations and affect the performance of texture generation. The method would be much stronger with an end-to-end trained human body model, as suggested by R3.

**2. Comparison from multiple views:** Fig. 1 illustrates multiple views of our rendered images on Market-1501 and DeepFashion. It can be seen that the details are still preserved well even for unseen parts of input images. We will show more qualitative results and comparison with RSTG [33] from multiple views in the future version.

**3. Comparison to PIFu:** To be general, our method does not use any ground-truth 3D textures during training. This is more applicable, however, limited by this it is more challenging. Therefore, they are along different developments and it is unfair to compare our method with PIFu, which requires ground-truth 3D texture supervision.

**To R1 & R3: Novelty and differences w.r.t. RSTG [33]:** Different from RSTG [33], we explicitly encourage the cross-view consistency during training. Besides, we also propose to use semantic parsing of human image as the model input to reduce the appearance variation and preserve pose information of human body. To our knowledge, these have not been explored before in the task of texture generation.

**To R1: 1. It's unclear if there is significant improvement over RSTG [33] from Fig. 5:** Compared to RSTG [33], our method can keep more details of input images, e.g., T-shirts in Row 1-2 and shorts in Row 3 in Fig. 5 (a). Our method also achieves much better quantitative results in terms of SSIM and mask-SSIM, as reported in Table 2.

**2. Using LPIPS for evaluation:** We evaluate our method with LPIPS on Market-1501. For our method, the average distance computed by the LPIPS metric is 2.440, and for RSTG [33], the one is 2.685. Thus, our method still outperforms RSTG in terms of LPIPS.

**3. CMR has much higher mask-SSIM score than RSTG:** Although CMR looks worse from a global perspective, it does better than RSTG in some local details. This may cause that CMR obtains higher mask-SSIM score which focuses on the area of human body.

**4. Using the disjoint UV mapping:** Good suggestion. Actually, although we only use the simple spherical UV mapping, our method still achieves better performance than RSTG [33] which already uses the disjoint UV mapping, especially in detail preserving. This further demonstrates the effectiveness of the proposed components.

**5. Missing relevant citations:** Thanks for your suggestion. We will include them in the updated version.

**To R2: 1. The texture copying is a little "randomly" when the texture is complicated (Fig.1 (b)):** Since we learn to predict the texture flow without ground-truth 3D texture supervision, it is impossible to perfectly match the pixel locations between the mesh surface and input image. When the texture pattern is complicated, such dislocation will be magnified visually and the texture appears to be copied "randomly".

**2. The texture tends to have front-back symmetry in Fig.1:** You are right. Since only one image is used as input and ground-truth 3D textures are unavailable during training, it is very hard to infer the accurate texture of the back (front) view from the front (back) view, even for humans. Thus, the model tends to simply copy the predicted texture flow. Further improvements could be possible with prior knowledge about front-back texture relations learned from data.

**To R3: 1. Relation with [A]:** Both the task and method are different. [A] is a nice work aiming at learning a powerful representation for 3D pose estimation, where a spatial transform based bidirectional novel view synthesis is further proposed to exploit view consistency. In contrast, our task is to generate a physical 3D model for computer graphics system, with a parameterized 3D mesh along with detailed textures on it. This explicit 3D modeling is different from spatial transform. Thanks for your suggestion. We will add related discussion in the updated version.

**2. Could the existing model be trained end-to-end, including pose and shape estimation:** Yes, it could be end-to-end trained if annotations of human joints and masks are given. In such way, more accurate 3D poses and shapes can be estimated to further improve the performance of texture generation, and at the same time the computation can be reduced by sharing the same backbone network. Thanks for the suggestion as a nice future work.

**3. What happens when the pose or shape estimation is inaccurate:** The perceptual loss can mitigate the impact of inaccurate pose or shape estimation to a certain extent, but inaccurate human parsing may affect the performance of texture flow prediction.

**4. Why is the head excluded in DeepFashion:** We exclude the head to make visualization focus on the body texture generation, which is mainly considered in this paper.

[Meta-Review · NeurIPS 2020]

Following the author response and the online discussion, concerns were resolved and all reviewers recommend the paper for acceptance. The authors are asked to address the reviewers' comments in the final version.